# Tardive Syndrome Is a Mysterious Phenomenon with Different Clinical Manifestations—Review

**DOI:** 10.3390/jcm12041498

**Published:** 2023-02-14

**Authors:** Samih Badarny, Rima Nassar, Yazid Badarny

**Affiliations:** 1Neurology Department, Galilee Medical Center, Nahariya 221001, Israel; 2Bar Ilan Faculty of Medicine, Zafed 1311502, Israel; 3Faculty of Medicine, Israel Technion, Haifa 3109601, Israel; 4Neurosurgery Department, Rambam Medical Center, Haifa 3109601, Israel

**Keywords:** tardive syndrome, physiopathology, clinical manifestation, management

## Abstract

Tardive syndrome (TS) refers to persistent hyperkinetic, hypokinetic, and sensory complaints appearing after chronic neuroleptics and other dopamine receptor-blocking agents (DRBAs). It is defined as involuntary movements, often rhythmic, choreiform, or athetoid, involving the tongue, face, extremities, and sensory urges such as akathisia and lasts for a few weeks. TS develops in association with neuroleptic medication usage for a few months at least. There is usually a delay between the initiation of the causative drug and the onset of abnormal movements. However, it was soon noted that TS can also develop early, even days or weeks after DRBAs begin. However, the longer the exposure, the greater the risk of developing TS. Tardive dyskinesia, dystonia, akathisia, tremor, and parkinsonism are frequent phenomenologies of this syndrome.

## 1. Introduction

Tardive syndrome is seen in psychiatric patients undergoing neuroleptic treatment and patients without psychiatric diseases related to other dopamine receptor-blocking agents(DRBAs) for different causes or related to other agents.In both situations, it can bring embarrassment, disability, deformity, adverse healthcare outcomes, and diminished quality of life [1]. The diagnostic and statistical Manual of Mental Disorders and DSM V define TS as a DRBA-induced movement disorder that persists despite discontinuation or change in medication dosage. Symptoms must persist for a month or more after discontinuing medication to confirm TS diagnosis [2]. There are two components of TS, the first is the abnormal movements or sensory symptoms related to DRBA exposure and the second is the persisting or worsening symptoms after the offending drug is withdrawn. Establishing a TS diagnosis after three months of DRBA use with more than 8 weeks of symptoms is acceptable. TS is distinguishable from similar symptoms provoked as acute side effects of using DRBA, which disappear shortly after the drug is stopped [3]. In 1964, Faurbye et al. proposed the term tardive dyskinesia to refer to the delayed onset of these movements. Previously, Cohen, Shonecker and Sigwald, and Faurbie described similar cases without naming them tardive syndrome or dyskinesia [4,5,6,7]. The main hypothesis of TS is related to D2 or even D3 upregulation and hypersensitivity of dopaminergic receptors or increased synaptic numbers and morphological changes in the synapses [8,9]. However, D2 hypersensitivity alone is insufficient to explain TS, particularly why only a fraction of patients develop TS despite animal models indicating that D2 hypersensitivity is present in all exposed animals, even those without abnormal movements described in the model [10]. Although it can be related to drug type, a deficient adaptation of 5-HT2A or NMDA receptors or can be related to vesicular monoamine transporter type 2 [11,12]. The oxidative stress hypothesis can explain TS as blocking dopamine receptors and increasing dopamine metabolism, resulting in free radical production that influences the basal ganglia, striatum, and substantia nigra and manifests TS. Li et al. [13]. found that schizophrenics with TD had significantly reduced gray matter volumes in the bilateral inferior and right superior frontal gyri in correlation with symptom severity. Sarro et al. [14], however, found that TD-related volume reductions were predominantly subcortical, involving the basal ganglia and thalamus. Drugs that do not have DRBA, such as tricyclics, SSRI, antiepileptics, antihistamines, anticholinergics, and amphetamines, are thought to produce tardive dyskinesia and cause signs and symptoms identical to TS, but there is no consensus [15] (Table 1).

## 2. Clinical Manifestation

Tardive syndrome (TS) is the spectrum of all persistent hyperkinetic, hypokinetic, and sensory phenomenologies resulting from chronic exposure to DRBA or other agents. They include dystonia, chrea, tremor, akathisia, parkinsonism, tics and myoclonus. The first known type is tardive dyskinesia; however, we summarize different forms of the syndrome below (Table 2).

### 2.1. Tardive Dyskinesia

Orobucolingual dyskinesia was described as the first type of TS and called tardive dyskinesia (TD). TD presents with slow, repetitive, stereotypic writhing and a protruding tongue, puckering lips, lip smacking, and chewing movements, with other choreic–athetoid movements of the trunk, such as rocking back and forth, thrusting movements of the pelvis, piano playing motion of hands, hyperventilation, irregular breathing, postural myoclonus, and motor and phonic tics [16,17]. Although orobucolingual involuntary movements can affect speech, eating is usually not impaired because involuntary movements cease when an object touches the lips—a sensory trick [3]. TD is distinguishable from withdrawal emergent syndrome, in which symptoms begin after abruptly discontinuing DRBA but are transient and resolve within several weeks [18]. Other differential diagnoses of TD include the stereotypical movements of schizophrenia, senility, advanced age, or dental placement and prosthesis [19].

### 2.2. Tardive Dystonia

Young males tend to have more tardive dystonia than females and the elderly. The syndrome was first reported in 1982 by Burke et al. [20]. The predominant forms are cervical dystonia, blepharospasm, and jaw opening and closing dystonia. Cervical dystonia has retrocollis compared to idiopathic dystonia, which has latero or antero posturing with the trunk bent forward. In tardive dystonia, opisthotonic posturing commonly occurs when the patient is walking and is associated with the trunk facing backward. The arms are typically adducted and pronated, and the wrists are flexed. At symptoms onset, the dystonia is mostly focal, and the neck is the first to be affected. However, after months or years, most of the body progresses to segmental or generalized dystonia. Blepharospasm and jaw opening and closing dystonia do not differ from other idiopathic forms. Tardive dystonia is sometimes associated with TD or akathisia, which helps make differential diagnoses from idiopathic dystonia. Tardive dystonia can improve in children but less in adults. However, ceasing the drug and replacing it with a newer generation of antipsychotic drugs or providing dopamine depletors is the best order to improve the situation [21].

### 2.3. Tardive Akathisia

Aktasia is difficulty staying still, seated, or lying down, accompanied by the urge to move, creating a feeling of restlessness. It is often difficult for patients to describe, presenting as a subjective clinical complaint with semipurposeful or purposeless movements of the limbs. Crossing, abducting, adducting legs, sitting and standing alternately, and hand movements resembling the motion of a bird’s wings are the frequent movements of tardive akathisia [22]. This phenomenon is generally underdiagnosed and is a serious problem because it can lead to many adverse outcomes, such as poor compliance with medications, exacerbation of psychiatric symptoms, and in some cases, violence or suicide. Aktasia usually appears after long-term exposure to DRBA or during its reduction or ceasing; however, it can sometimes manifest after short exposure to these drugs, making it difficult to distinguish it from acute akathisia [23]. Mouth, tongue, and genital pain are considered a part of tardive akathisia, sometimes called tardive pain, as first reported by Hierholzer in 1989. A few more cases with this phenomenon have been described, and in most of them, the pain preoccupies the patient to the point of obsession [24,25].

### 2.4. Tardive Tremor

Stacy and Jankovic et al. described tardive tremor in 1992 and later published a case series of 10 patients [26,27,28]. It is distinct from the resting tremor of parkinsonism in that fewer postural, action, and rest tremors occur after long periods of neuroleptic therapy, and it is not well known and is often misdiagnosed [29]. Clinically similar to essential tremor, tardive tremor is usually not present before the administration of neuroleptic drugs and do not disappear. It becomes aggravated after ceasing the drug and does not respond to medicines for managing essential tremor. However, they respond to dopamine depletors or atypical neuroleptic drugs such as clozapine.

### 2.5. Tardive Parkinsonism

The incidence of parkinsonism is secondary to neuroleptics since tardive parkinsonism has decreased due to the use of atypical, new-generation neuroleptic drugs with less affinity for D2 receptors. Parkinsonism is found more in the elderly, females, and a long history of using DRBA. Ceasing the drug, unlike other forms of TS, improves parkinsonian symptoms within a few months or years. However, in about a quarter of cases, symptoms do not disappear but persist or even progress over time [30]. Usually, symptoms are quite symmetrical without rest tremor and no non-motor signs such as RBD, hyposmia, or autonomic symptoms compared to idiopathic Parkinson’s disease. They have also been shown to have a normal flourodopa PET scan compared to the idiopathic form [31,32,33].

## 3. Treatment of Tardive Syndrome

TS treatment is not simple and requires a strategy with a good treatment plan. Firstly, we need to recognize the offending drug, determine whether it is a typical or atypical neuroleptic drug, and determine if the patient needs it. First of all, you need to know the drug that causes the symptoms of TS, is it a typical or atypical neuroleptic drug?

As a principle, it is mandatory to involve the psychiatrist and raise the basic question: Is it possible to stop the antipsychotic drug? If not, replacing typical with atypical antipsychotic drugs should be suggested. Quetiapine and clozapine are recommended as antipsychotic replacement treatments, and they rarely, if ever, cause extrapyramidal disorders. However, if these two drugs fail to treat the underlying (psychiatric) disease, we recommend switching to aripiprazole, although it is not certain whether this strategy is widely used [34,35]. Tetrabenazine (TBZ) has been used in the United Kingdom for treating hyperkinetic movement disorders since 1971. It was approved in the USA in 2008 for Huntington.TBZ is a selective vesicular monoamine transporter inhibitor type 2, which reduces dopamine storage in vesicles and reduces dopamine stimulation. There are no double-blind, placebo-controlled studies for TBZ. However, two large retrospective reports on TBZ in TS showed that 85% of patients improved as either moderate or marked. The common side effects of TBZ are sedation, parkinsonism, depression, and insomnia, which are dose-related and decrease with dose reduction [36,37]. The short half-life of TBZ requires frequent dosing, which should start at low doses with careful titration and monitoring of side effects [38]. For these reasons, TBZ has become less common and relatively limited in recent years. However, its derivatives, such as deutetrabenazine and valbenazine, which have a longer half-life and fewer side effects, became more widely used than TBZ. These two drugs are probably more effective for dyskinesias, stereotypes, and akathisia than dystonia. Bhidayasiri et al. published a systematic review of practical treatment for these drugs, showing significant reductions in abnormal TS movements with no depression or parkinsonism in their 6–12 week study with open-label extensions [39] (Table 3).

Clonazepam as benzodiazepine was evaluated in a 12-week, double-blind, randomized, crossover trial of 19 chronically ill TS patients receiving DRBAs. Tardive symptoms reduced nearly a third from baseline after 12 weeks [40]. Diazepam and alprazolam use is limited and has not been proven to help TS. Amantadine is useful and effective in treating parkinsonian symptoms after exposure to DRBA. Levetiracitam has been shown to successfully improve TS in several case reports and open studies, and in one double-blind, placebo-controlled, randomized, 12-week, parallel group study has been reported in 50 subjects who were enrolled, 25 randomized to each group [41]. Piracetam, is a drug similar to levetiracitam used for myoclonus in Europe, was initially shown to improve TS symptoms [42].

Botulinum toxin type A for TS was described in several open case studies with some improvement and there is one small single blind study which showed benefit [43]. For tardive dystonia, such as cervical dystonia, oromandibular dystonia and blepharospasm, the botulinum toxin has been widely utilized. Several open-label retrospective reports support its use and the response of tardive dystonia is similar to that of idiopathic dystonia [44].

Deep brain stimulation of the globus pallidus interna (GPi-DBS) has been reported to treat TS, particularly tardive dystonia and classical tardive dyskinesia refractory to medical therapy. A blinded study of 19 patients receiving GPi-DBS for combined tardive dystonia/dyskinesia with 50% improvement that sustained as long as 11 years after surgery [45]. In another double-blind, sham stimulation–controlled trial of GPi-DBS tardive dystonia/dyskinesia in 25 patients. At 6 months of stimulation, both groups improved by 42% along with significant improvement in quality of life. DBS of the STN is less used, however Deng et.al reported well improvements in 10 patients that sustained for several years [46].

## 4. Conclusions

The purpose of our review was to collect data on an important issue that lacks absolute solutions and persists despite a new generation of antipsychotic drugs. TS is a syndrome secondary to DRBA exposure that has various forms; for example, TS is a syndrome secondary to exposure to DRBA that has various forms such as dyskinesia, dystonia, akathisia, tremors, and parkinsonism. These symptoms do not disappear when treatment ceases and may even worsen. Apart from discontinuing treatment or replacing it with atypical antipsychotic drugs such as quetiapine or clozapine, the therapeutic approach involves using dopamine depletors such as tetrabenazine or its derivatives, including deutetrabenazine and valbenazine, which improve symptoms with less side effects, however deep brain stimulation of GPi also has been reported to treat TS. Added to this that, DBS of GPi was also found to be effective if the medicines did not improve the TS symptoms.

## Figures and Tables

**Table 1 jcm-12-01498-t001:** Medications associated with tardive dyskinesia.

Neuroleptic agents
Anticholinergic agents
SSRI antidepressants as fluoxetine and sertraline
Other antidepressants as trazadone, clomipramine, amitriptyline
MAOI
Lithium
Antiemetics as metoclopramide, prochlorperazine
Anticonvulsants as carbamazepine and lamotrigine
Antihistamine as hydroxyzine
Anxiolytics as GABA agonist and barbiturates
Antimalarial as chloroquine

**Table 2 jcm-12-01498-t002:** Clinical manifestation of tardive syndrome.

Signs and Symptoms	Type of Dyskinesia
Oro-bucco-lingual chreoathetotic movements; sometimes tics or myoclonus	Tardive dyskinesia
Cervical dystonia; blepharospasm; jaw opening and closing	Tardive dystonia
dystonia	
Difficulty staying still while seated or lying down; urge to move mouth and tongue; genital pain	Tardive akathisia
Postural and actional tremor and less rest tremor; respond to dopamine depletors or clozapine	Tardive tremor
Symmetrical symptoms without rest tremor; non-motor signs such as RBD and hyposmia and autonomic symptoms.	Tardive parkinsonism

**Table 3 jcm-12-01498-t003:** Comparison of vesicular monoamine transporter type 2 inhibitors.

Characteristic	Tetrabenazine	Deutetrabenazine	Valbenazine
Mechanism	VMAT2 inhibitor	VMAT2 inhibitor	VMAT2 inhibitor
Half-life	5–7 h	9–10 h	15–22 h
Dose range	12.5–100 mg/d	6–48 mg/d	40–80 mg/d
Safety data	>50 y	>5 y	>5 y
Adverse events	1. Sedation 2. Parkinsonism 3. Depression	1. Sedation 2. Insomnia	1. Sedation 2. Headache 3. Fatigue

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
