# Peer review of "Tardive Syndrome Is a Mysterious Phenomenon with Different Clinical Manifestations—Review"

_jcm, 2023, doi:10.3390/jcm12041498_

Round 1

Reviewer 1 Report

This is a clinically relevant topic. 

The manuscript needs extensive editing by a native English speaker.

The authors have discussed the various TS, possible pathophysiology. A table mentioning the various drugs causing these syndromes could be added .Also  some mention about functional MRI  studies would be worthwhile. 

Author Response

the manuscript underwent english corrections

table of drug agents provoking Tardive syndrome wsa added

f MRI in Tardive dyskinesia was added

table 1

Table  1

Neuroleptic agents

Anticholinergic agents

SSRI antidepressants as fluoxetine and sertraline

Other antidepressants as trazadone, clomipramine, amitriptyline

MAOI

Lithium

 Antiemetics as metoclopramide, prochlorperazine

Anticonvulsants as carbamazepine and lamotrigine

Antihistamine as hydroxyzine

Anxiolytics as GABA agonist and barbiturates

Antimalarial  as chloroquine

 Medications associated with tardive dyskinesia

Reviewer 2 Report

The authors' attempts are good, but there is very little content.

The rationale for the classification of Tardive Syndrome is poor. The description of each symptom is also poor. There is also too little information on treatment, and not enough on surgical treatment such as DBS.

Author Response

i made english corrections

the rational information was improved

more inormation on treatment  

Round 2

Reviewer 2 Report

The description of treatment is still scant. In particular, botulinum treatment and surgical treatment need to be added.

Author Response

hello sir

  1. I checked the references and were added several
  2. My revision was marked up using TRACK CHANGES  
  3. I am the the last version to the journal and to you 
  4. I hope that i completed your notes
  5. Sincerely yours  samih badarny 
